# Cucurbitacin E Suppresses Adipogenesis and Lipid Accumulation in 3T3-L1 Adipocytes Without Cytotoxicity

**DOI:** 10.3390/biomedicines13081826

**Published:** 2025-07-25

**Authors:** Tien-Chou Soong, Kuan-Ting Lee, Yi-Chiang Hsu, Tai-Hsin Tsai

**Affiliations:** 1Department of Weight Loss and Health Management Center, E-DA Dachang Hospital and I-Shou University, Kaohsiung 807, Taiwan; obesitysurgery@gmail.com; 2Department of Asia Obesity Medical Research Center, E-DA Cancer Hospital and I-Shou University, Kaohsiung 824, Taiwan; 3Graduate Institutes of Medicine, College of Medicine, Kaohsiung Medical University, Kaohsiung 807, Taiwan; ayta860404@gmail.com; 4School of Medicine, I-Shou University, Kaohsiung 824, Taiwan; jenway74@isu.edu.tw; 5Division of Neurosurgery, Department of Surgery, Kaohsiung Medical University Hospital, Kaohsiung 807, Taiwan; 6Department of Surgery, School of Medicine, College of Medicine, Kaohsiung Medical University, Kaohsiung 807, Taiwan

**Keywords:** cucurbitacin E, 3T3-L1 cells, adipocyte differentiation, obesity, Anti-Obesity Medication (AOM)

## Abstract

**Background**: Cucurbitacin E (CuE), a natural tetracyclic triterpenoid compound extracted from the melon stems of Cucurbitaceae plants, has been reported to exhibit anti-inflammatory and anti-cancer properties, along with the ability to enhance cellular immunity. However, its role and molecular mechanism in regulating lipid metabolism and adipogenesis remain unclear. This study aims to investigate the potential anti-adipogenic and anti-obesity effects of CuE in 3T3-L1 adipocytes. **Materials and Methods**: 3T3-L1 preadipocytes were cultured and induced to differentiate using a standard adipogenic cocktail containing dexamethasone, 3-isobutyl-1-methylxanthine (IBMX), and insulin (DMI). CuE was administered during the differentiation process at various concentrations. Lipid accumulation was assessed using Oil Red O staining, and cell viability was evaluated via the MTT assay. To determine whether CuE induced apoptosis or necrosis, flow cytometry was performed using annexin V/PI staining. Additional molecular analyses, such as Western blotting and RT-PCR, were used to examine the expression of key adipogenic markers. **Results**: Treatment with CuE significantly reduced lipid droplet formation in DMI-induced 3T3-L1 adipocytes in a dose-dependent manner, as shown by decreased Oil Red O staining. Importantly, CuE did not induce apoptosis or necrosis in 3T3-L1 cells at effective concentrations, indicating its safety toward normal adipocytes. Moreover, CuE treatment downregulated the expression of adipogenic markers such as PPARγ and C/EBPα at both mRNA and protein levels. **Discussion**: Our findings suggest that CuE exerts a non-cytotoxic inhibitory effect on adipocyte differentiation and lipid accumulation. This anti-adipogenic effect is likely mediated through the suppression of key transcription factors involved in adipogenesis. The absence of cytotoxicity supports the potential application of CuE as a safe bioactive compound for obesity management. Further investigation is warranted to elucidate the upstream signaling pathways and in vivo efficacy of CuE. **Conclusions**: Cucurbitacin E effectively inhibits adipogenesis in 3T3-L1 adipocytes without inducing cytotoxic effects, making it a promising candidate for the development of functional foods or therapeutic agents aimed at preventing or treating obesity. This study provides new insights into the molecular basis of CuE’s anti-obesity action and highlights its potential as a natural lipogenesis inhibitor.

## 1. Introduction

Obesity is related to a variety of diseases, including hypertension, dyslipidemia, type 2 diabetes, and fatty liver [1]. Obesity is caused by an imbalance between energy intake and consumption, which promotes lipogenesis and the enlargement of adipocytes, leading to the growth of adipose tissue [2,3]. It has also been reported that central obesity is associated with resistance to peripheral insulin action [4,5,6] (2024).

Fat cell counts remain relatively stable throughout adulthood. A recent report showed that long-term high-fat diet (HFD) intake has been shown to preferentially initiate lipogenesis in white adipose tissue, suggesting that lipogenesis is associated with obesity and obesity-related chronic diseases. White adipocyte tissue is formed from adipocytes [7,8,9]. The increase in fat cell tissue mass in the body is due to lipogenesis from preadipocytes. Typically, the mouse adipocyte line 3T3-L1 is used to evaluate adipogenesis and adipocyte differentiation in vitro. Therefore, many studies will study the adipogenesis mechanism and anti-obesity effect by inducing 3T3-L1 cell differentiation through treatment with insulin, dexamethasone, 3-isobutyl-1-methylxanthine, dexamethasone, and rosiglitazone [10,11,12]. Adipogenesis is a two-step process in which mesenchymal precursors of the adipocyte system limit the formation of progenitor cells, and the formed cells differentiate into insulin-sensitive mature adipocytes [13]. During the differentiation stage, preadipocytes undergo early and terminal differentiation through growth arrest, mitotic clonal expansion, and the regulation of transcription factors and adipogenesis-related genes, thereby promoting adipocyte development [14,15]. Currently, pancreatic lipase inhibitors are the only long-term treatments for obesity [16]. In addition, today’s obesity drugs also exhibit dangerous side effects [17]. Therefore, new drug targets need to be developed.

Members of the cucurbitacin family, including the cucumber, melon, watermelon, pumpkin, and squash, have been found to possess a range of biological and pharmacological activities [18]. About 20 kinds of cucurbitacin compounds have been identified based on structural changes. Cucurbitacin compounds, which are representative components of these Cucurbitaceae plants, mainly produce a bitter taste and are therefore exploited as a defense mechanism [19]. Current evidence indicates that Cucs has growth inhibitory effects on several cancer cells such as bladder cancer, pancreatic cancer, hepatocarcinoma, breast cancer, and leukemia [20,21,22,23,24].

However, the role of CuE in reducing obesity and related metabolic complications in 3T3-L1 has not been studied [25]. Therefore, we hope that through this research, cucurbitacin E can become an auxiliary drug to treat and inhibit lipogenesis in 3T3-L1 cells in the future. This study focuses on exploring the mechanism by which CuE inhibits the growth and adipogenesis of the 3T3-L1 cell line. It is expected that CuE can take advantage of the rapid growth characteristics of 3T3-L1 cells to exert an inhibitory effect on adipocytes. It does not cause apoptosis or necrosis. According to this study, it has the effect of inhibiting fat growth and lipogenesis.

## 2. Materials and Methods

### 2.1. Cell Lines and Reagents

The 3T3-L1 preadipocyte cell line, originally obtained from the American Type Culture Collection (ATCC; catalog number ATCC-CL-173), was used for all in vitro experiments in this study. The cells were routinely maintained in standard tissue culture dishes under sterile conditions. The culture medium consisted of high-glucose Dulbecco’s Modified Eagle Medium (DMEM) supplemented with 2 mM L-glutamine, which supports optimal cell growth and metabolic activity. Additionally, the medium contained 1.5 g/L sodium bicarbonate to help regulate the pH balance in the culture environment. To promote cell proliferation and maintain cell health, the DMEM was further supplemented with 10% (*v*/*v*) fetal bovine serum (FBS), a source of essential growth factors, and a 1% (*v*/*v*) penicillin–streptomycin solution to prevent bacterial contamination. The cells were incubated at 37 °C in a humidified atmosphere composed of 5% carbon dioxide (CO_2_), which is standard for mammalian cell culture. The medium was replaced every 2–3 days to ensure nutrient availability and maintain optimal culture conditions throughout the experimental period.

### 2.2. Cell Differentiation Culture

The 3T3-L1 preadipocytes were seeded into 6 cm cell culture dishes at an initial density of approximately 3 × 10^5^ cells/mL, using Dulbecco’s Modified Eagle Medium (DMEM) supplemented with 10% fetal bovine serum (FBS) as the growth medium. The cells were allowed to proliferate under standard culture conditions (37 °C, 5% CO_2_, and saturated humidity) until they reached full confluency, forming a uniform monolayer across the culture surface. Upon reaching this stage, differentiation was initiated by replacing the growth medium with a well-established adipogenic induction medium, commonly referred to as MDI. This medium consists of DMEM supplemented with 10% FBS and a cocktail of differentiation-inducing agents: 10 μg/mL insulin (Sigma, St. Louis, MO, USA), 1 μM dexamethasone (Sigma, USA), and 0.5 mM 3-isobutyl-1-methylxanthine (IBMX; Sigma, USA). The cells were maintained in this induction medium for a period of two days under the same incubator conditions.

After the initial 48 h of induction, the MDI medium was replaced with DMEM supplemented with 10% FBS and 10 μg/mL insulin alone. This insulin-containing maintenance medium was renewed every two days to support continued adipogenic differentiation. Over the course of approximately six to eight days, noticeable morphological changes were observed: the cells gradually transformed from their original fibroblast-like, spindle-shaped appearance into a more rounded or spherical shape. Concurrently, small intracellular lipid droplets began to accumulate within the cytoplasm, a hallmark of terminal adipocyte differentiation. These lipid-laden cells were recognized as mature adipocytes, thereby confirming the successful induction of adipogenesis in the 3T3-L1 cell model.

### 2.3. Oil Red O Staining

Following the completion of the adipogenic differentiation process, the accumulation of intracellular lipid droplets in the 3T3-L1 adipocytes was assessed using Oil Red O staining, a well-established histological method for detecting neutral lipids. Oil Red O is a lipid-soluble diazo dye that selectively stains triglycerides and lipids within the cytoplasm, rendering the lipid droplets visible as vivid red inclusions under light microscopy [26].

Prior to staining, the culture medium was carefully aspirated, and the cells were gently rinsed with phosphate-buffered saline (PBS) to remove residual serum and metabolites. The cells were then fixed with 10% neutral buffered formaldehyde at room temperature for 30 min, allowing for the preservation of cellular morphology and lipid integrity. After fixation, the formaldehyde solution was removed, and the cells were rinsed thoroughly with sterile distilled water to eliminate any remaining fixative. Subsequently, freshly prepared and filtered Oil Red O working solution was applied to each well, ensuring complete coverage of the cell monolayer. The staining was performed in the dark at room temperature for 30 min, allowing the dye to penetrate and stain the intracellular lipid droplets.

After incubation, the excess dye was discarded, and the cells were washed multiple times with sterile water until the background appeared clear, ensuring the removal of unbound stain. The stained cells were then observed under an inverted light microscope, and representative images were captured for qualitative documentation of lipid accumulation. To quantify the lipid content, a fixed volume of isopropanol was added to each stained sample to elute the retained Oil Red O dye from the lipid droplets. The optical density (OD) of the extracted dye solution was subsequently measured at a wavelength of 510 nm using a microplate spectrophotometer (Powerwave XS, Bio-Tek Instruments, Winooski, VT, USA). The absorbance readings directly correlate with the intracellular lipid content, providing a semi-quantitative assessment of adipogenesis.

### 2.4. Cell Viability

To assess the cytotoxicity and potential inhibitory effects of CuE on adipocyte viability, a colorimetric MTT assay was employed using 3T3-L1 preadipocytes [27]. Approximately 3 × 10^3^ 3T3-L1 cells per well were seeded into sterile 96-well tissue culture plates, and the cells were allowed to adhere overnight under standard conditions (37 °C, 5% CO_2_, and a humidified atmosphere). On the following day, the culture medium was replaced with fresh DMEM supplemented with 10% fetal bovine serum (FBS), and experimental groups were treated with varying concentrations of CuE (Sigma-Aldrich, USA) dissolved in DMSO, while the final DMSO concentration was kept constant and below cytotoxic thresholds. The control group received only the vehicle (DMSO) in DMEM containing 10% FBS and was not exposed to CuE.

Cells were incubated with CuE for 24 h, after which the MTT reagent (3-[4,5-dimethylthiazol-2-yl]-2,5-diphenyltetrazolium bromide; MERCK, Darmstadt, Germany) was added to each well at a final concentration of 0.5 mg/mL. Plates were returned to the incubator and maintained under the same conditions (37 °C and 5% CO_2_) for an additional 4 h to allow viable cells to reduce the MTT dye into insoluble formazan crystals. Following incubation, the supernatant was carefully aspirated, and dimethyl sulfoxide (DMSO; MERCK) was added to each well to fully dissolve the formazan crystals, ensuring homogeneous color development.

The absorbance of each well was measured at 540 nm using a microplate reader (PowerWave XS, BioTek Instruments, USA). The intensity of absorbance was directly proportional to the number of metabolically active, viable cells. All treatments were performed in triplicate, and results were expressed as the percentage of cell viability relative to the untreated control group. This assay enabled a quantitative assessment of CuE’s effect on 3T3-L1 cell viability under adipogenic conditions.

### 2.5. Cell Cycle Analysis

To evaluate the effects of CuE on the regulation of cell cycle progression in human 3T3-L1 adipocytes cell lines, we employed flow cytometric analysis using propidium iodide (PI) staining, a widely accepted method for quantifying DNA content and identifying cell cycle distribution. Human 3T3-L1 adipocytes cells were seeded into 6 cm tissue culture dishes at an initial density of 3 × 10^5^ cells/mL and cultured in Dulbecco’s Modified Eagle Medium (DMEM) supplemented with 10% FBS. The cells were incubated under standard culture conditions (37 °C, 5% CO_2_, and a humidified environment) for 24 h to allow adherence and stabilization prior to drug treatment.

After the initial incubation, cells were treated with a range of concentrations of Cucurbitacin E, which was dissolved in DMSO and diluted in the fresh culture medium. The control group was treated with the same concentration of the DMSO vehicle but without the addition of CuE. Following a 24 h treatment period, both treated and control cells were harvested by trypsinization and washed twice with cold phosphate-buffered saline (PBS). The collected cell pellets were fixed overnight in ice-cold 70% ethanol at −20 °C for at least 24 h, a critical step to ensure the permeabilization of the nuclear membrane and the preservation of DNA content for downstream analysis.

After fixation, the cells were centrifuged to remove ethanol and washed with PBS to eliminate any residual fixative. The cell pellets were then resuspended in propidium iodide (PI) staining buffer, which included 0.02 mL of 1 mg/mL PI (BD Biosciences, Franklin Lakes, NJ, USA), 0.02 mL of 5% Triton X-100 (Sigma, USA), and 0.01 mL of 2 mg/mL RNase A (Sigma, USA), all diluted in 0.95 mL of PBS for a final reaction volume of 1 mL per sample. This mixture allows PI to intercalate into the cellular DNA, while RNase A ensures the removal of RNA contamination, thus enhancing the accuracy of DNA quantification. The samples were incubated at room temperature for 30 min in the dark to protect the photosensitive PI from degradation.

Following staining, the prepared samples were analyzed using a BD FACSCalibur flow cytometer (Becton Dickinson, Franklin Lakes, NJ, USA). For each sample, a minimum of 10,000 events were acquired to ensure robust statistical representation. The acquired data were analyzed using the WinMDI 2.8 software, which provided graphical output and a quantitative evaluation of the percentage of cells in each phase of the cell cycle (G0/G1, S, and G2/M phases). This allowed for a detailed comparison of cell cycle alterations between CuE-treated and untreated groups, offering insight into the potential of CuE to induce cell cycle arrest in human lipoma cells.

### 2.6. Apoptosis and Necrosis Analysis

To investigate whether CuE induces apoptotic or necrotic cell death in human lipoma cells, we employed a flow cytometry-based assay using the Annexin V-FITC Apoptosis Detection Kit (Strong Biotech, New Taipei, Taiwan), which enables the distinction between early apoptosis, late apoptosis, and necrosis [28]. This method takes advantage of Annexin V’s affinity for phosphatidylserine residues that become externalized on the outer leaflet of the plasma membrane during the early stages of apoptosis, as well as the use of PI to identify the loss of membrane integrity typically associated with late apoptosis or necrosis. Human lipoma cells were seeded in 6 cm tissue culture dishes at a density of 1 × 10^5^ cells/mL using Dulbecco’s Modified Eagle Medium (DMEM) supplemented with 10% FB. The cells were maintained in a humidified incubator at 37 °C with 5% CO_2_ and allowed to adhere and stabilize for 24 h prior to treatment. After this initial incubation period, the cells were treated with various concentrations of CuE, which was prepared in DMSO and diluted in fresh DMEM medium to ensure that the final DMSO concentration remained below cytotoxic thresholds.

After 4 h of exposure to CuE, the cells were gently harvested and washed twice with ice-cold PBS to remove the residual medium and CuE. The cell pellets were then resuspended in binding buffer provided by the Annexin V-FITC kit and incubated with Annexin V-FITC and PI according to the manufacturer’s protocol. The staining procedure was conducted in the dark at room temperature for 15 min, ensuring the protection of the fluorescent dyes from photobleaching and maintaining assay sensitivity. Following staining, samples were immediately analyzed using a BD FACSCalibur flow cytometer (Becton Dickinson, USA). For each treatment condition, a minimum of 10,000 cells (1 × 10^4^ events) were acquired to allow for statistically meaningful analysis. The collected data were processed and interpreted using the WinMDI 2.8 software, which allowed for the generation of dot plots that distinguish viable, Apoptotic (Annexin V+/PI+) cell populations. This analytical approach enabled the quantitative assessment of CuE-induced cytotoxicity and provided a reliable method to determine whether CuE treatment promotes apoptosis or necrosis in lipoma cells. The results derived from this assay contribute to the understanding of CuE’s mechanistic role in regulating lipoma cell survival and its potential as a therapeutic agent with minimal toxicity to normal adipocytes.

### 2.7. Statistical Analysis

All quantitative data obtained from experimental assays were expressed as the mean ± the standard error of the mean (SEM) and were derived from a minimum of three independent experiments conducted under identical conditions to ensure reproducibility and statistical reliability. For statistical comparisons between the control and treatment groups, a Student’s t-test was performed. Where applicable, post hoc analyses were conducted to further validate pairwise group differences. Statistical significance was defined as a *p*-value less than 0.05 (*p* < 0.05), indicating that the observed differences were unlikely to have occurred by random chance. All statistical analyses were performed using the appropriate software tools to ensure rigor and precision in the interpretation of the results.

## 3. Results

### 3.1. Preparation and Induction of Differentiation in 3T3-L1 Cells

To initiate adipogenic differentiation, murine 3T3-L1 preadipocytes were cultured and subsequently induced to differentiate into mature adipocytes using a well-established hormonal cocktail. Throughout the differentiation process, a gradual increase in cell size was observed, corresponding to the accumulation of intracellular lipid droplets, a hallmark of adipogenesis. This morphological transformation was visually monitored and documented to ensure consistent progression toward full differentiation. As shown in Figure 1A, 3T3-L1 cells exhibited progressive morphological changes characteristic of mature adipocytes, including increased cell volume and the appearance of multiple lipid vacuoles within the cytoplasm. To further confirm adipocyte maturation, Oil Red O (ORO) staining was performed on differentiated cells. The ORO assay, which specifically stains neutral lipids, revealed a marked enhancement in lipid droplet accumulation in cells that underwent adipogenic induction compared to undifferentiated controls. The substantial increase in lipid staining intensity and droplet density validated the successful differentiation of 3T3-L1 preadipocytes into mature adipocytes under the applied induction conditions.

These results confirmed the effectiveness of the differentiation protocol and established a reliable in vitro model for subsequent experiments evaluating the anti-adipogenic effects of cucurbitacin E.

### 3.2. Effect of CuE on the Growth and Cell Viability of 3T3-L1 Cells

To determine whether CuE influences the survival and proliferation of preadipocytes, we hypothesized that CuE may suppress the viability of 3T3-L1 cells in a dose-dependent manner. To test this, 3T3-L1 cells were cultured in vitro and treated with various concentrations of CuE for 24 h. Cell viability was assessed using the MTT assay, which quantifies mitochondrial enzymatic activity as a proxy for viable, metabolically active cells. The results revealed a clear inhibitory trend: as the CuE concentration increased, cell viability decreased accordingly. Specifically, higher doses of CuE resulted in a significant reduction in MTT absorbance, indicating decreased cellular proliferation and survival (Figure 1B). These findings suggest that CuE exerts a cytostatic or cytotoxic effect on 3T3-L1 cells, potentially interfering with their progression toward adipogenic differentiation.

Therefore, we propose that CuE may serve as a negative regulator of preadipocyte growth and could play a role in modulating adipogenesis through the suppression of early-stage cell proliferation.

### 3.3. Effect of CuE on Apoptosis and Cell Necrosis in 3T3-L1 Cells

To investigate whether CuE induces apoptotic or necrotic cell death in 3T3-L1 preadipocytes, we performed flow cytometric analysis using the ApopNexin FITC Apoptosis Detection Kit (Chemicon, Moses Lake, DC, USA). 3T3-L1 cells were treated with varying concentrations of CuE for 4 h, followed by double staining with Annexin V-FITC and PI. This dual-staining approach allowed us to distinguish viable cells (Annexin V^−^/PI^−^) from early apoptotic (Annexin V^+^/PI^−^), late apoptotic (Annexin V^+^/PI^+^), and necrotic cells (Annexin V^−^/PI^+^), enabling a more precise assessment of CuE’s effects on cell death pathways.

As shown in Figure 2A, representative flow cytometry plots demonstrated that treatment with CuE at different concentrations did not result in a marked increase in apoptosis or necrosis compared to untreated controls. Furthermore, quantitative analysis (Figure 2B) revealed that even at higher CuE concentrations (2.5, 5, and 10 nM), there was no statistically significant elevation in the proportion of apoptotic cells. These results indicate that CuE does not trigger programmed cell death or membrane-compromising necrosis in 3T3-L1 cells under the experimental conditions applied.

Taken together, our findings suggest that CuE does not impair 3T3-L1 cell viability via apoptotic or necrotic mechanisms and that its previously observed inhibitory effects on cell proliferation likely occur through alternative, non-apoptotic pathways.

### 3.4. Effect of CuE on Cell Cycle in 3T3-L1 Cells

To investigate whether CuE exerts its anti-proliferative effects through the modulation of the cell cycle, we analyzed the cell cycle distribution of 3T3-L1 preadipocytes following CuE treatment using flow cytometry. The goal of this experiment was to determine whether CuE induces cell cycle arrest at specific phases, thereby influencing cell proliferation. 3T3-L1 cells were treated with increasing concentrations of CuE (0, 2.5, 5, and 10 nM) for 24 h, stained with propidium iodide, and subjected to flow cytometric analysis to quantify the proportion of cells in the G0/G1, S, and G2/M phases.

As shown in Figure 3A, CuE treatment did not result in any significant changes in the distribution of cells in different cell cycle phases. At all tested CuE concentrations, the percentage of cells in the G0/G1 phase, S phase, and G2/M phase remained essentially unchanged, and only CuE was found to significantly cause a classic cell cycle arrest in the S phase in 3T3-L1 cells. In addition, as shown in Figure 3B, even after 24 h of CuE treatment, no significant aggregation of cells in any specific phase was observed, which further supports the conclusion that CuE does not exert its anti-proliferative effect by blocking the cell cycle.

These results imply that the growth-inhibitory properties of CuE in 3T3-L1 cells are likely independent of cell cycle arrest mechanisms. Instead, alternative regulatory pathways—potentially involving differentiation signaling, cytoskeletal disruption, or metabolic modulation—may account for the observed suppression of cell proliferation.

## 4. Discussion

In this study, the successful establishment of a reliable 3T3-L1 preadipocyte differentiation model provided a robust in vitro platform for evaluating the anti-adipogenic effects of CuE. Utilizing this system, our findings demonstrate that CuE functions as a negative regulator of preadipocyte proliferation and may exert anti-adipogenic effects by interfering with early-stage cell growth. Importantly, CuE did not compromise cell viability through apoptotic or necrotic mechanisms, suggesting that its inhibitory effects on cell proliferation are mediated through non-apoptotic pathways. Moreover, the suppression of cell growth by CuE appears to be independent of cell cycle arrest, implicating alternative mechanisms—such as interference with adipogenic signaling pathways, cytoskeletal dynamics, or cellular metabolic processes—in mediating its anti-proliferative activity. These results collectively point to a multifaceted role of CuE in modulating adipocyte development, warranting further investigation into its underlying molecular targets [29,30,31,32,33,34].

Over the past few hundred years, bioactive compounds from medicinal plants have emerged as alternatives to traditional obesity treatments. As potential candidates for the development of anti-obesity drugs, the molecular mechanisms of action of medicinal plant compounds are being studied in depth [20,35]. In this study, we target adipose tissue biology and dysfunction to treat metabolic syndrome. Increasing evidence shows that Cucs and CuE, in particular, possess anti-proliferative, anti-inflammatory, and autophagic properties [36,37]. However, the role of cucurbitacins in adipose tissue biology is unclear [38]. This study provides the first evidence of CuE-mediated inhibition of adipocytes in 3T3L1 mice.

CuE, a naturally occurring triterpenoid compound derived from Cucurbitaceae plants, has been reported to inhibit adipogenesis. CuE exerts its anti-adipogenic effects primarily by modulating key signaling pathways involved in adipocyte differentiation. Specifically, CuE downregulates the expression of pivotal adipogenic transcription factors, such as peroxisome proliferator-activated receptor gamma (PPARγ) and CCAAT/enhancer-binding protein alpha (C/EBPα), which are essential for the maturation of preadipocytes into adipocytes [25,39]. Additionally, CuE has been shown to activate the AMP-activated protein kinase (AMPK) pathway, leading to enhanced energy metabolism and the suppression of lipogenic gene expression. Another potential mechanism involves the activation of the Wnt/β-catenin signaling pathway [39], which inhibits early stages of adipocyte differentiation by interfering with the induction of PPARγ and C/EBPα [38]. Notably, our laboratory’s recent experiments using 3T3-L1 preadipocyte cells demonstrated that low-dose CuE treatment effectively suppresses adipogenesis. This suppression is accompanied by a significant downregulation of adipogenic gene expression, aligning with previously reported findings. Collectively, these results suggest that CuE inhibits adipocyte differentiation through multiple molecular mechanisms and may serve as a potential therapeutic agent for obesity diseases.

CuE, a naturally occurring triterpenoid compound derived from Cucurbitaceae plants, has been reported to inhibit lipogenesis. CuE exerts its anti-lipogenic effects primarily by modulating key signaling pathways involved in lipid synthesis [38]. Specifically, CuE downregulates the expression of pivotal lipogenic transcription factors, such as sterol regulatory element-binding protein 1c (SREBP-1c), fatty acid synthase (FAS), and acetyl-CoA carboxylase (ACC), which are essential for the synthesis of fatty acids and triglycerides [40]. Additionally, CuE has been shown to activate the AMPK pathway, leading to enhanced energy metabolism and the suppression of lipogenic gene expression. AMPK activation results in the phosphorylation and inhibition of ACC, thereby reducing lipid accumulation within cells [38]. Notably, our laboratory’s recent experiments using 3T3-L1 preadipocyte cells demonstrated that CuE treatment effectively suppresses lipogenesis. This suppression is accompanied by a significant downregulation of lipogenic gene expression, aligning with previously reported findings. Collectively, these results suggest that CuE inhibits lipid synthesis through multiple molecular mechanisms and may serve as a potential therapeutic agent for metabolic disorders such as non-alcoholic fatty liver disease (NAFLD) and hyperlipidemia.

## 5. Conclusions

In conclusion, this study successfully validated a robust in vitro adipogenesis model using 3T3-L1 cells, providing a reliable platform to investigate the anti-adipogenic properties of CuE. Our findings demonstrate that CuE acts as a negative regulator of preadipocyte proliferation, thereby potentially modulating adipogenesis at early stages. Notably, the observed growth-inhibitory effects of CuE were not associated with apoptosis, necrosis, or cell cycle arrest, suggesting that CuE exerts its anti-proliferative activity through alternative, non-canonical mechanisms. These mechanisms may involve interference with adipogenic signaling pathways, cytoskeletal organization, or metabolic processes, highlighting CuE as a promising compound for further exploration in adipogenesis and obesity-related research.

## Figures and Tables

**Figure 1 biomedicines-13-01826-f001:**
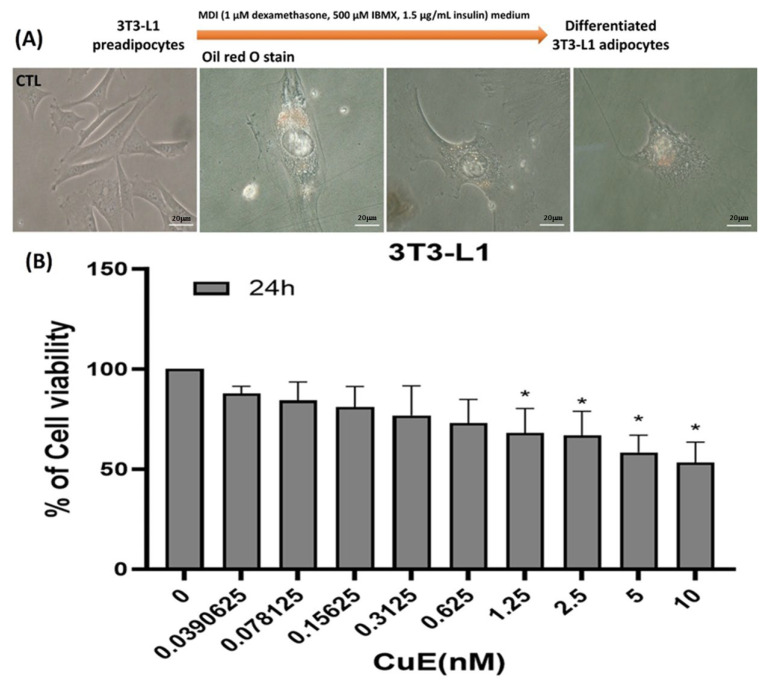
(**A**) Promotion of adipogenesis in vitro by 3T3-L1 cells. Differentiation of 3T3-L1 preadipocytes into mature adipocytes was evaluated using ORO staining, which specifically stains intracellular lipid droplets. The control group (CTL) consisted of undifferentiated 3T3-L1 preadipocytes cultured under standard growth conditions. Scale bar = 20 μm. (**B**) Effects of CuE on 3T3-L1 cell viability in a dose-dependent manner. To assess the cytotoxic effects of CuE, 3T3-L1 preadipocytes were treated with increasing concentrations of CuE for 24 h, and cell viability was measured using the MTT assay. The results demonstrate a dose-dependent decrease in cell viability following CuE treatment, indicating that higher concentrations of CuE significantly impair the growth and metabolic activity of 3T3-L1 cells. Data are presented as the mean ± SEM from three independent experiments (n = 3 per group). * *p* < 0.01 compared to the untreated control group was considered statistically significant. These findings suggest that CuE may act as a potential suppressor of preadipocyte proliferation. CuE, cucurbitacin E.

**Figure 2 biomedicines-13-01826-f002:**
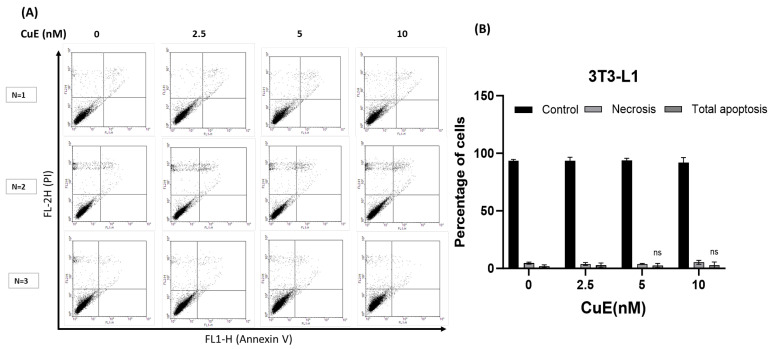
Effect of CuE on apoptosis and necrosis in 3T3-L1 preadipocytes. (**A**) Representative dot plots show the distribution of apoptotic and necrotic populations across different CuE treatment groups. No substantial increase in apoptotic or necrotic cells was observed in the CuE-treated groups compared to the untreated control. (**B**) The quantification of total apoptotic cells (early + late apoptosis) revealed no statistically significant increase at any tested CuE concentration, indicating that CuE does not trigger apoptosis in 3T3-L1 cells under these conditions. Data represent the mean ± SEM from three independent experiments (n = 3 per group) compared to the 0 nM CuE control. These results suggest that the inhibitory effect of CuE on cell proliferation is not mediated by apoptosis or necrosis. CuE, cucurbitacin E; PI, propidium iodide. ns: not significant.

**Figure 3 biomedicines-13-01826-f003:**
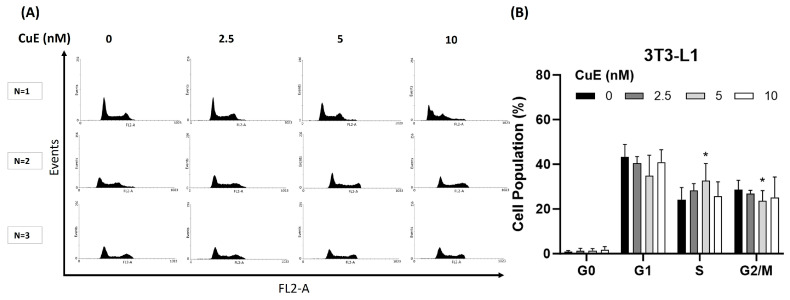
Effect of CuE on cell cycle progression in 3T3-L1 preadipocytes. (**A**) 3T3-L1 cells were treated with different concentrations of CuE (0, 2.5, 5, and 10 nM) for 24 h. Cells were then fixed, stained with propidium iodide (PI), and subjected to flow cytometry to determine the DNA content and analyze their distribution across cell cycle phases (G0/G1, S, and G2/M). Representative histograms illustrate the DNA content profiles for each treatment group. (**B**) Quantitative analysis showed that CuE treatment significantly altered the rise in S phase and the percentage of cells in any specific phase of the cell cycle compared to the control group. The proportion of cells in G0/G1 phase, S phase, and G2/M phase remained unchanged at all CuE concentrations, indicating that CuE did not induce cell cycle arrest in 3T3-L1 cells under the tested conditions. Data are shown as the mean ± SEM from three independent experiments (n = 3 per group). ns: not significant. These findings suggest that the anti-proliferative effect of CuE on 3T3-L1 cells is not mediated through the modulation of cell cycle progression. CuE, cucurbitacin E; PI, propidium iodide (* *p* < 0.05).

## Data Availability

The data used to support this study are available upon request from the corresponding authors.

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
