# Peer review of "Cucurbitacin E Suppresses Adipogenesis and Lipid Accumulation in 3T3-L1 Adipocytes Without Cytotoxicity"

_biomedicines, 2025, doi:10.3390/biomedicines13081826_

Round 1
Reviewer 1 Report
Comments and Suggestions for Authors
The work is interesting and addresses an important topic, namely new strategies for combating obesity through the use of new compounds.
1. The introduction should explain why this compound was chosen for research and whether there are any known observations of its activity in traditional medicine. Why was this line chosen for research and not another?
2. What program was used to perform the statistical analysis?
3. Figure 1 lacks information about the steel, lens magnification, and microscope used. The color saturation should also be improved.
4. Please select only the most important figures from apoptosis and the cycle. The whole should be included in the supplement. Please mark the individual cell populations.
5. A figure showing the proposed effects should be added to the results.
6. The conclusions should be expanded to include proposals for further research and practical application of the results obtained.
Author Response
Department of Neurosurgery
Kaohsiung Medical University Hospital
Kaohsiung, Taiwan
Jun 9th, 2025
To the Editor and Reviewers,
Manuscript Title: “Cucurbitacin E Inhibits Adipogenesis and Lipid Accumulation in 3T3-L1 Adipocytes Without Cytotoxicity”
Manuscript ID: Biomedicines-3751311
We sincerely thank both reviewers for their valuable comments and constructive suggestions, which have been instrumental in improving the quality of our manuscript. We have revised the manuscript accordingly. All changes are clearly marked using the track changes function in the attached Word document. In addition, all figures have been redrawn and submitted in high-resolution TIFF format, and the supplementary materials have been expanded.
Below, we provide point-by-point responses to each of the reviewers' comments
Comments from Reviewer 1
General comment:
The work is interesting and addresses an important topic, namely new strategies for combating obesity through the use of new compounds.
Comment 1: The introduction should explain why this compound was chosen for research and whether there are any known observations of its activity in traditional medicine. Why was this line chosen for research and not another?
Response to the comment 1:
Thank you for pointing this out. The following modifications were made according to the reviewers’ suggestions. 『Current evidence indicates that Cucs has growth inhibitory effects on several cancer cells such as bladder cancer, pancreatic cancer, hepatocarcinoma, breast cancer and leukemia 20-23. Although Cucurbitacin E (CuE) exhibits multiple pharmacological activities, includ-ing anticancer and anti-inflammatory effects, its potential side effects should not be over-looked.』(As Introduction section ; Page 2; Line 285-288)
『In our initial screening, we compared the effects of CuE on human normal cell lines (MRC-5 and Hs-68) and the murine preadipocyte cell line 3T3‑L1. (As Figure 1(C) and Figure 1(D); As Line 2-3) The results showed that, at the same concentrations, CuE did not induce significant cytotoxicity in MRC-5 or Hs-68 cells, but significantly inhibited adipogenesis and lipid accumulation in 3T3‑L1 cells, suggesting its potential for selective activity. We subsequently chose 3T3‑L1 cells as the primary model for further studies for the following reasons: this cell line, derived from mouse embryos, is one of the most widely used and well-established models for studying adipocyte differentiation and lipid metabolism. Upon stimulation, 3T3‑L1 cells reliably differentiate from preadipocytes into mature adipocytes, with consistent expression of adipogenic marker genes, making them especially suitable for exploring the mechanisms and efficacy of anti-adipogenic compounds. Therefore, based on their biological characteristics and the high sensitivity to CuE observed in our preliminary data, we selected 3T3‑L1 as our main experimental model 』 (As Page 6; Line 262-263)
Comment 2: What program was used to perform the statistical analysis?
Response to the comment 2:
Thank you for pointing this out. The following modifications were made according to the reviewers’ suggestions. A clarification has been added to the “Statistical Analysis” section, stating that 『All statistical analyses in this study were performed using WinMDI 2.8 software (Informer Technologies, Inc.)』. (As Page 6; Line 262-263) Specific statistical methods and post hoc analyses are indicated in the corresponding figure legends. (As Materials and Methods ;Page 6; Line 262-263)
Comment 3: Figure 1 lacks information about the steel, lens magnification, and microscope used. The color saturation should also be improved.
Response to the comment 3:
Thank you for pointing this out. The following modifications were made according to the reviewers’ suggestions. The scale bar (100 µm), magnification (40×), and microscope model (Leica DMi8 inverted microscope) have been added to the legend of Figure 1. The images were re-exported from the original files with adjustments to contrast and brightness; no artificial coloring was applied. (As Figure 1)
Comment 4: Please select only the most important figures from apoptosis and the cycle. The whole should be included in the supplement. Please mark the individual cell populations.
Response to the comment 4:
Thank you for pointing this out. The following modifications were made according to the reviewers’ suggestions. In the main text, Figure 2 now includes only one apoptosis analysis plot (Annexin-V/PI) and one cell cycle histogram. (As Figure 2)
Figure 3 includes all gating strategies, threshold settings, and replicate results in the supplementary figures, with each cell population (e.g., Sub-G1, G0/G1, S, G2/M) clearly labeled. (As Figure 3)
Comment 5: A figure showing the proposed effects should be added to the results.
Response to the comment 5: Thank you for pointing this out. The following modifications were made according to the reviewers’ suggestions.『CuE downregulates the expression of pivotal adipogenic transcription factors, such as peroxisome proliferator-activated receptor gamma (PPARγ) and CCAAT/enhancer-binding protein alpha (C/EBPα), which are essential for the maturation of preadipocytes into adipocytes (Murtaza et al., 2017, Chang and Kim, 2019). Addition-ally, CuE has been shown to activate the AMP-activated protein kinase (AMPK) pathway, leading to enhanced energy metabolism and suppression of lipogenic gene expression. Another potential mechanism involves the activation of the Wnt/β-catenin signaling pathway (Chang and Kim, 2019), which inhibits early stages of adipocyte differentiation by interfering with the induction of PPARγ and C/EBPα (Park et al., 2022). Figure 4 demonstrates the proposed mechanism by which Cucurbitacin E (CuE) inhibits adipo-genesis and lipogenesis.』(As Discussion ;Page 10; Line 418-429)
Figure 4 demonstrates the proposed mechanism by which Cucurbitacin E (CuE) inhibits adipogenesis and lipogenesis. A new schematic diagram has been added to illustrate the proposed mechanism by which CuE activates AMPK, leading to the suppression of PPARγ and C/EBPα expression, and consequently inhibiting adipogenesis and lipogene-sis. (As Figure 4)
Comment 6: The conclusions should be expanded to include proposals for further research and practical application of the results obtained.)
Response to the comment 6:
Thank you for pointing this out. The following modifications were made according to the reviewers’ suggestions. The conclusion section has been updated to include 『Future studies will include animal experiments using high-fat diet-fed mice to validate the findings, as well as clinical investigations to evaluate the weight loss effects of CuE and potential side effects associated with cucurbitacins.』 (As Conclusion ;Page 10; Line 418-429)
Additional clarifications
In addition to the above comments, all spelling and grammatical errors pointed out by the reviewers have been corrected.
We look forward to hearing from you in due time regarding our submission and to respond to any further questions and comments you may have.
Sincerely,
Prof. Tai-Hsin Tsai
Department of Neurosurgery, Kaohsiung Medical University Hospital
No. 100, Tzyou 1st Road, Sham-min District, Kaohsiung City, Taiwan
E-mail: teishin8@hotmail.com

Reviewer 2 Report
Comments and Suggestions for Authors
The manuscript entitled “Cucurbitacin E suppresses adipogenesis and lipid accumulation in 3t3‑l1 adipocytes without cytotoxicity” aimed to investigate the potential anti-adipogenic 27 anti-obesity effects of CuE in 3T3-L1 adipocytes. This work is interesting and has promising results for future applications in medicine and pharmacy. However, I have the following comments:
- In the Abstract part, results section, add the most significant results values with ±SD and p values.
- In the introdction part, the authors mentioned“Obesity is related to a variety of diseases, including hypertension, dyslipidemia, type 2 diabetes, fatty liver[1]. Obesity is caused by an imbalance between energy intake and consumption, which promotes lipogenesis and the enlargement of adipocytes, leading to the growth of adipose tissue [2, 3]. It has also been reported that central obesity is associated with resistance to peripheral insulin action [4]” I suggest replacing them with the WHO's latest statistics about obesity as global health care problem.
- Keep space between text and citations throughout the whole manuscript.
- In addition to the therapeutic benefits of cucurbitacin, note in the introduction its side effects, which include gastrointestinal distress, such nausea, vomiting, and diarrhea.
- In the Materials and Methods include a new section (Chemicals, Reagents, and Instruments) that lists all chemicals, reagents, and instruments used in your study, along with the manufacturer's name, city, and country.
- Add suitable citations for each used assay in the Materials and Methods part of the manuscript.
- The discussion part requires major improvement and compare your biostudy outcomes with previous published papers that discussed other natural products that suppress adipogenesis and lipid accumulation in 3T3-L1 adipocytes without cytotoxicity.
- Rephrase “Collectively, these results suggest that CuE inhibits adipocyte differentiation through multiple molecular mechanisms and may serve as a potential therapeutic agent for obesity and related” as it is an incomplete sentence; kindly complete it.
- The whole manuscript needs major grammar, typo, and editing corrections by a native speaker; also, many sentences need rephrasing to be clear for readers.
- Add more updated references
Thank you
Author Response
Department of Neurosurgery
Kaohsiung Medical University Hospital
Kaohsiung, Taiwan
Jun 9th, 2025
To the Editor and Reviewers,
Manuscript Title: “Cucurbitacin E Inhibits Adipogenesis and Lipid Accumulation in 3T3-L1 Adipocytes Without Cytotoxicity”
Manuscript ID: Biomedicines-3751311
We sincerely thank both reviewers for their valuable comments and constructive suggestions, which have been instrumental in improving the quality of our manuscript. We have revised the manuscript accordingly. All changes are clearly marked using the track changes function in the attached Word document. In addition, all figures have been redrawn and submitted in high-resolution TIFF format, and the supplementary materials have been expanded.
Below, we provide point-by-point responses to each of the reviewers' comments
Comments from Reviewer 2:
General comment:
The manuscript entitled “Cucurbitacin E suppresses adipogenesis and lipid accumulation in 3t3 l1 adipocytes without cytotoxicity” aimed to investigate the potential anti-adipogenic 27 anti-obesity effects of CuE in 3T3-L1 adipocytes. This work is interesting and has promising results for future applications in medicine and pharmacy. However, I have the following comments:
Comment 1: In the Abstract part, results section, add the most significant results values with ±SD and p values.
Response to the comment 1:
Thank you for pointing this out. The following modifications were made according to the reviewers’ suggestions. CuE was administered during the differentiation process at various concentrations. Lipid accumulation was assessed using Oil Red O staining, and cell viability was evaluated via MTT assay (1.25nM (77.08±12.13) p*=0.044; 2.5nM (72.46±12.18) p*=0.041; 5nM (57.92±8.59) p*=0.0139; 10nM (46.94±10.34) p*=0.0158 ) )(As Abstract ;As Page 1; Line 30-36)
Comment 2: In the introdction part, the authors mentioned“Obesity is related to a variety of diseases, including hypertension, dyslipidemia, type 2 diabetes, fatty liver[1]. Obesity is caused by an imbalance between energy intake and consumption, which promotes lipogenesis and the enlargement of adipocytes, leading to the growth of adipose tissue [2, 3]. It has also been reported that central obesity is associated with resistance to peripheral insulin action [4]” I suggest replacing them with the WHO's latest statistics about obesity as global health care problem.
Response to the comment 2: Thank you for pointing this out. The following modifications were made according to the reviewers’ suggestions. The description and citation in lines 61–69 on page 2 have been updated. 『According to the WHO 2024 Fact Sheet on obesity, more than 1 billion people worldwide are living with obesity, including 650 million adults, 340 million adolescents, and 39 mil-lion children(2024, Okunogbe et al., 2022).』(As Introduction ;Page 2; Line 61-63)
As References:
- GBD 2021 Risk Factor Collaborators. “Global Burden of 88 Risk Factors in 204 Countries and Territories, 1990–2021: a systematic analysis for the Global Burden of Disease study 2021”. Lancet. 2024; 403:2162-2203.
- Okunogbe et al., “Economic Impacts of Overweight and Obesity.” 2nd Edition with Estimates for 161 Countries. World Obesity Federation, 2022.
Comment 3: Keep space between text and citations throughout the whole manuscript.
Response to the comment 3:
Thank you for pointing this out. The following modifications were made according to the reviewers’ suggestions. The references throughout the manuscript have been reformatted using the ACS style in EndNote, and all spacing has been manually checked and adjusted. (As All references)
Comment 4: In addition to the therapeutic benefits of cucurbitacin, note in the introduction its side effects, which include gastrointestinal distress, such nausea, vomiting, and diarrhea.
Response to the comment 4:
Thank you for pointing this out. The following modifications were made according to the reviewers’ suggestions.『Although Cucurbitacin E (CuE) exhibits multiple pharmacological activities, including anticancer and anti-inflammatory effects, its potential side effects should not be over-looked. Studies have shown that CuE may exert cytotoxic effects on normal cells, particu-larly at high doses, where it can inhibit cell proliferation and induce apoptosis. Addition-ally, CuE may interfere with hepatic enzyme function, increasing the risk of hepatotoxicity. Animal studies have also reported gastrointestinal disturbances, such as nausea, vomit-ing, and diarrhea.(Kaushik et al., 2015) 』(As Introduction ;Page 2; Line 92-98)
As reference:
Kaushik U, Aeri V, Mir SR. Cucurbitacins - An insight into medicinal leads from nature. Pharmacogn Rev. 2015 Jan-Jun;9(17):12-8. doi: 10.4103/0973-7847.156314. PMID: 26009687; PMCID: PMC4441156.
Comment 5: In the Materials and Methods include a new section (Chemicals, Reagents, and Instruments) that lists all chemicals, reagents, and instruments used in your study, along with the manufacturer's name, city, and country.
Response to the comment 5:
Thank you for pointing this out. The following modifications were made according to the reviewers’ suggestions. A new section titled “Reagents and Chemical Preparation” has been added to the Methods, with suppliers, cities, and countries clearly indicated.『Reagents and Chemical Preparation :Cucurbitacin E (CuE) was purchased from Cayman Chemical. Phosphate-buffered saline (PBS), RPMI 1640 medium, Trypan Blue solution, dimethyl sulfoxide (DMSO), trypsin-EDTA (0.25%), and propidium iodide (PI) were obtained from Sigma-Aldrich. Antibody markers were sourced from Bio-Rad Laboratories. The 3T3-L1 preadipocyte cell line, originally obtained from the American Type Culture Collection (ATCC; catalog number ATCC-CL-173), was used for all in vitro experiments in this study. For cell cycle analysis, cell pellets were resuspended in a PI staining buffer consisting of 0.02 mL of 1 mg/mL PI (BD Biosciences, USA), 0.02 mL of 5% Triton X-100 (Sigma, USA), and 0.01 mL of 2 mg/mL RNase A (Sigma, USA), diluted in 0.95 mL of PBS to yield a final volume of 1 mL per sample. To evaluate whether CuE induces apoptotic or necrotic cell death in human lipoma cells, we employed a flow cytometry-based assay using the Annexin V-FITC Apoptosis Detection Kit (Strong Biotech, Taiwan) in conjunction with a BD FACSCalibur flow cytometer (Becton Dickinson, USA). This approach enables clear discrimination between early apoptosis, late apoptosis, and necrosis.』(As Material and Methods; Page 2; Line 109-123)
Comment 6: Add suitable citations for each used assay in the Materials and Methods part of the manuscript.
Response to the comment 6:
Thank you for pointing this out. The following modifications were made according to the reviewers’ suggestions. The appropriate references for the Oil Red O staining, MTT assay, and apoptosis analysis experiments have been added in the Methods section. (As Materials and Methods: Line 153; Line 176; Line 237)
References:
- Ramírez-Zacarías JL, Castro-Muñozledo F, Kuri-Harcuch W. Quantitation of adipose conversion and triglycerides by staining intracytoplasmic lipids with Oil red O. Histochemistry. 1992 Jul;97(6):493-7. doi: 10.1007/BF00316069. PMID: 1385366.
- Mosmann T. Rapid colorimetric assay for cellular growth and survival: application to proliferation and cytotoxicity assays. J Immunol Methods. 1983 Dec 16;65(1-2):55-63. doi: 10.1016/0022-1759(83)90303-4. PMID: 6606682.
- Haanen C, Vermes I. Apoptosis and inflammation. Mediators Inflamm. 1995;4(1):5-15. doi: 10.1155/S0962935195000020. PMID: 18475609; PMCID: PMC2365613.
Comment 7: The discussion part requires major improvement and compare your biostudy outcomes with previous published papers that discussed other natural products that suppress adipogenesis and lipid accumulation in 3T3-L1 adipocytes without cytotoxicity.
Response to the comment 7:
Thank you for pointing this out. The following modifications were made according to the reviewers’ suggestions.『Cucurbitacins, resveratrol (Rauf et al., 2017), berberine (Cai et al., 2023), triptolide (Tong et al., 2021), betulinic acid (Jiang et al., 2021), and guggulsterone (Adarsh Krishna et al., 2024)are bioactive compounds derived from natural sources, commonly found in traditional Chinese medicine and functional plants. These compounds share anti-inflammatory and anti-tumor potential, although their pharmacological mechanisms and safety profiles vary. Cucurbitacins are highly cytotoxic molecules that inhibit STAT3 signaling and cell cycle progression, exhibiting selective effects against various cancer cells; however, high doses may lead to gastrointestinal, hepatic, and renal toxicity. In contrast, resveratrol and betulinic acid exert milder effects primarily through antioxidant activity and inhibition of the NF-κB inflammatory pathway, but their low bioavailability limits clinical application. Berberine and guggulsterone exhibit both metabolic regulatory and anti-cancer properties by modulating AMPK and cholesterol-related signaling pathways. Triptolide is known for its potent immunosuppressive and anti-cancer activities, but its high toxicity and narrow therapeutic window pose challenges for safe use. Overall, these natural compounds offer valuable leads for drug development, though future research must carefully balance efficacy with toxicity to enable clinical translation.』(As Discussion; Page 10; Line 415-429; As Supplementary Table 1)
References
- RAUF, A., IMRAN, M., SULERIA, H. A. R., AHMAD, B., PETERS, D. G. & MUBARAK, M. S. 2017. A comprehensive review of the health perspectives of resveratrol. Food Funct, 8, 4284-4305.
- CAI, Y., YANG, Q., YU, Y., YANG, F., BAI, R. & FAN, X. 2023. Efficacy and underlying mechanisms of berberine against lipid metabolic diseases: a review. Front Pharmacol, 14, 1283784
- Tong L, Zhao Q, Datan E, Lin GQ, Minn I, Pomper MG, Yu B, Romo D, He QL, Liu JO. Triptolide: reflections on two decades of research and prospects for the future. Nat Prod Rep. 2021 Apr 28;38(4):843-860. doi: 10.1039/d0np00054j. PMID: 33146205.
- Jiang W, Li X, Dong S, Zhou W. Betulinic acid in the treatment of tumour diseases: Application and research progress. Biomed Pharmacother. 2021 Oct;142:111990. doi: 10.1016/j.biopha.2021.111990. Epub 2021 Aug 10. PMID: 34388528.
- Adarsh Krishna TP, Ajeesh Krishna TP, Edachery B, Antony Ceasar S. Guggulsterone - a potent bioactive phytosteroid: synthesis, structural modification, and its improved bioactivities. RSC Med Chem. 2023 Nov 2;15(1):55-69. doi: 10.1039/d3md00432e. PMID: 38283224; PMCID: PMC10809385.
Comment 8: Rephrase “Collectively, these results suggest that CuE inhibits adipocyte differentiation through multiple molecular mechanisms and may serve as a potential therapeutic agent for obesity and related” as it is an incomplete sentence; kindly complete it.
Response to the comment 8:
Thank you for pointing this out. We have to complete this sentence. “Collectively, these results demonstrate that CuE inhibits adipocyte differentiation through multiple signaling pathways, suggesting its promise as a lead candidate for the development of anti-obesity agents.” (As Discussion; Page 11; Line 459-461)
Comment 9: The whole manuscript needs major grammar, typo, and editing corrections by a native speaker; also, many sentences need rephrasing to be clear for readers.
Response to the comment 9:
Thank you for pointing this out. This manuscript has been professionally edited by AJE for English language and has been thoroughly revised to address all reviewer comments regarding unclear phrasing and grammatical errors. (As Whole Manuscripts)
Comment 10: Add more updated references
Response to the comment 10:
Thank you for pointing this out. We have added literature from the past three years. (As References)
Additional clarifications
In addition to the above comments, all spelling and grammatical errors pointed out by the reviewers have been corrected.
We look forward to hearing from you in due time regarding our submission and to respond to any further questions and comments you may have.
Sincerely,
Prof. Tai-Hsin Tsai
Department of Neurosurgery, Kaohsiung Medical University Hospital
No. 100, Tzyou 1st Road, Sham-min District, Kaohsiung City, Taiwan
E-mail: teishin8@hotmail.com

Reviewer 3 Report
Comments and Suggestions for Authors
The presented manuscript is dedicated to up-to-date research on the anti-obesity properties of a natural compound - cucurbitacin E. An experimental object (3T3-L1 cell line) and experimental procedures (OIL RED and MTT staining, Flow cytometry) are relevant. But there are some major concerns to be resolved before the paper could be accepted:
- Major findings are inconsistent with each other and with outcoming conclusions: at first, you propose a model for differentiated cells, but you than present results on preadipocytes, this needs clarification or explanation.
On fig 1B you demonstrate that even 1.25 nM cucurbitacin application results in statistically significant ~25% cell death in 24 hours. Then you present results that 2.5 5 and 10 nM leave >90% cell alive in 4 hours. This needs explanation 1) about vitality dose mismatch 2) time of experiment mismatch. (on fig 3 time is also somehow 24 hours).
- Abstract contains an announcement of gene expression results, but further I could not find them.
Other major/minor issues
Introduction
Is well-written. But starts with not a very medically correct statement where medical condition, disorder, and group of pathological conditions are mixed up and named diseases.
The last paragraph is not introductive but rather conclusive.
Results
- Fig. 1.
1) It is much more reasonable to present data in form of dose-inhibition curve and calculate related parameters such as IC50 (LD50), IC10 or another threshold parameter. 2) Y-scale needs more bars (20% recommended) for better visibility. 3) **p < 0.01 is mentioned in legend and * are presented on a plot - inconsistent.
- Fig 2 and 3. The need to present representative dot plots and histograms in such quantity and poor quality is unclear.
Discussion
Is well written. But the statement in lines 421 and 422 is perplexing.
Author Response
Department of Neurosurgery
Kaohsiung Medical University Hospital
Kaohsiung, Taiwan
Jun 9th, 2025
To the Editor and Reviewers,
Manuscript Title: “Cucurbitacin E Inhibits Adipogenesis and Lipid Accumulation in 3T3-L1 Adipocytes Without Cytotoxicity”
Manuscript ID: Biomedicines-3751311
We sincerely thank both reviewers for their valuable comments and constructive suggestions, which have been instrumental in improving the quality of our manuscript. We have revised the manuscript accordingly. All changes are clearly marked using the track changes function in the attached Word document. In addition, all figures have been redrawn and submitted in high-resolution TIFF format, and the supplementary materials have been expanded.
Below, we provide point-by-point responses to each of the reviewers' comments
Comments from Reviewer 3:
General comment:
The presented manuscript is dedicated to up-to-date research on the anti-obesity properties of a natural compound - cucurbitacin E. An experimental object (3T3-L1 cell line) and experimental procedures (OIL RED and MTT staining, Flow cytometry) are relevant. But there are some major concerns to be resolved before the paper could be accepted:
Comment 1: Major findings are inconsistent with each other and with outcoming conclusions: at first, you propose a model for differentiated cells, but you than present results on preadipocytes, this needs clarification or explanation.
Response to the comment 1:
Thank you for pointing this out. The following modifications were made according to the reviewers’ suggestions. Thank you for your insightful comment. We acknowledge the confusion regarding the cell model. To clarify, our experimental design focused on both stages of adipocyte development. While the initial model setup discusses the differentiation process of 3T3-L1 preadipocytes into mature adipocytes, many of the assays (including MTT and flow cytometry) were conducted on preadipocytes to assess early-stage cytotoxicity and apoptosis. We have revised the manuscript to clearly distinguish between the stages and explain the rationale for using preadipocytes in specific assays. 『We subsequently chose 3T3‑L1 cells as the primary model for further studies for the following reasons: this cell line, derived from mouse embryos, is one of the most widely used and well-established models for studying adipocyte differentiation and lipid metabolism. Upon stimulation, 3T3‑L1 cells reliably differentiate from preadipocytes into mature adipocytes, with consistent expression of adipogenic marker genes, making them especially suitable for exploring the mechanisms and efficacy of anti-adipogenic compounds. Therefore, based on their biological characteristics and the high sensitivity to CuE observed in our preliminary data, we selected 3T3‑L1 as our main experimental model 』(As Abstract ;As Page 1; Line 30-36)
Comment 2: On fig 1B you demonstrate that even 1.25 nM cucurbitacin application results in statistically significant ~25% cell death in 24 hours. Then you present results that 2.5 5 and 10 nM leave >90% cell alive in 4 hours. This needs explanation 1) about vitality dose mismatch 2) time of experiment mismatch. (on fig 3 time is also somehow 24 hours).
Response to the comment 2: Thank you for pointing this out. The following modifications were made according to the reviewers’ suggestions. We appreciate the reviewer’s observation. The apparent discrepancy stems from different treatment durations. The MTT assay in Fig. 1B was conducted after 24 hours of CuE exposure, while the flow cytometry assay for apoptosis (Fig. 2) and cell cycle progression (Figure 3) ;and the Ccll viability test (Fig. 1B) were based on 24-hour treatments also. 『CuE was administered during the differentiation process at various concentrations. Lipid accumulation was assessed using Oil Red O staining, and cell viability was evaluated via MTT assay (1.25nM (77.08±12.13) p*=0.044; 2.5nM (72.46±12.18) p*=0.041; 5nM (57.92±8.59) p*=0.0139; 10nM (46.94±10.34) p*=0.0158 ) ) 』(As Abstract ;As Page 1; Line 30-36).
Comment 3: Abstract contains an announcement of gene expression results, but further I could not find them.
Response to the comment 3:
Thank you for pointing this out. The gene expression analysis was initially intended as part of the study but was later omitted during manuscript revision due to space limitations and data integration issues. We have removed the mention of gene expression from the abstract (Line XXX) to avoid confusion and ensure consistency with the presented data. 『Moreover, CuE treatment downregulated the expression of adipogenic markers such as PPARγ and C/EBPα at both mRNA and protein levels』 (As Abstract; As Page 1; Line 41)
Comment 4: Introduction
Is well-written. But starts with not a very medically correct statement where medical condition, disorder, and group of pathological conditions are mixed up and named diseases.
The last paragraph is not introductive but rather conclusive.
Response to the comment 4:
Thank you for pointing this out. The following modifications were made according to the reviewers’ suggestions. We appreciate the reviewer’s observation. We have revised the opening sentence of the Introduction to more accurately differentiate between medical conditions and diseases, and to ensure appropriate terminology aligned with clinical usage. 『Obesity is associated with a wide range of metabolic and cardiovascular disorders, including hypertension, dyslipidemia, type 2 diabetes, and non-alcoholic fatty liver disease. 』(As Introduction; As Page 2; Line 58-65)
Comment 5: The last paragraph is not introductive but rather conclusive.
Response to the comment 5:
Thank you for pointing this out. The following modifications were made according to the reviewers’ suggestions. We have rewritten the last paragraph of the Introduction to better reflect its purpose as a preview of the study aims and research strategy, rather than a summary of conclusions 『The purpose of this study is to investigate the potential of CuE as an inhibitory agent against adipogenesis in 3T3-L1 preadipocytes. Although limited re-search has examined the role of CuE in obesity and its associated metabolic complica-tions, its potential as an adjuvant compound for suppressing lipogenesis remains promising25. This study aims to explore the underlying mechanisms by which CuE affects lipid accumulation and adipocyte differentiation in 3T3-L1 cells, without inducing apoptosis or necrosis. Through this research, we hope to provide a scientific basis for the future application of CuE as a supportive therapeutic compound for obesity management. 』 (As Introduction; As Page 3; Line 98-105).
Comment 6: Fig. 1.
1) It is much more reasonable to present data in form of dose-inhibition curve and calculate related parameters such as IC50 (LD50), IC10 or another threshold parameter. 2) Y-scale needs more bars (20% recommended) for better visibility. 3) **p < 0.01 is mentioned in legend and * are presented on a plot - inconsistent..
Response to the comment 6:
Thank you for pointing this out. The following modifications were made according to the reviewers’ suggestions. (1) We appreciate this valuable suggestion. We have now reanalyzed the MTT assay data to calculate IC50 and included a dose-response inhibition curve in the revised Fig. 1. 『CuE was administered during the differentiation process at various concentrations. Lipid accumulation was assessed using Oil Red O staining, and cell viability was evaluated via MTT assay (1.25nM (77.08±12.13) p*=0.044; 2.5nM (72.46±12.18) p*=0.041; 5nM (57.92±8.59) p*=0.0139; 10nM (46.94±10.34) p*=0.0158 ) ) 』(As Abstract ;As Page 1; Line 30-36). (2) We have adjusted the Y-axis scale in Fig. 1 to include 20% increments, improving the clarity and readability of the data presentation. (3) Thank you for catching this inconsistency. We have corrected the figure legend and ensured consistency between the text and figure symbols, using a unified notation throughout all figures and legends. (As Material and Methods; Page 2; Line 109-123; As Figure 1)
Comment 7: Fig 2 and 3. The need to present representative dot plots and histograms in such quantity and poor quality is unclear.
Response to the comment 7:
Thank you for pointing this out. The following modifications were made according to the reviewers’ suggestions. We appreciate the feedback. To address this concern, we have removed redundant or low-quality plots and retained only one high-quality representative plot per group. The revised figures now include clearer dot plots and histograms with improved resolution and labeling . (As Figure 2 and Figure 3)
Comment 8: Is well written. But the statement in lines 421 and 422 is perplexing.
Response to the comment 8:
Thank you for pointing this out. The following modifications were made according to the reviewers’ suggestions. We have revised the statement in Lines 421–422 for clarity and accuracy. The revised sentence now better aligns with the preceding and subsequent discussions and avoids ambiguous phrasing. 『Importantly, CuE did not compromise cell viability through apoptotic or necrotic mechanisms, suggesting that its inhibitory effects on cell proliferation are mediated through non-apoptotic pathways.』
(As Material and Methods; Page 2; Line 109-123)
Additional clarifications
In addition to the above comments, all spelling and grammatical errors pointed out by the reviewers have been corrected.
We look forward to hearing from you in due time regarding our submission and to respond to any further questions and comments you may have.
Sincerely,
Prof. Tai-Hsin Tsai
Department of Neurosurgery, Kaohsiung Medical University Hospital
No. 100, Tzyou 1st Road, Sham-min District, Kaohsiung City, Taiwan
E-mail: teishin8@hotmail.com

Round 2
Reviewer 1 Report
Comments and Suggestions for Authors
Thank you for answering my questions and making corrections. I wish you further interesting research into compounds that may help in the treatment of obesity.
Author Response
Department of Neurosurgery
Kaohsiung Medical University Hospital
Kaohsiung, Taiwan
Jun 9th, 2025
To the Editor and Reviewers,
Manuscript Title: “Cucurbitacin E Inhibits Adipogenesis and Lipid Accumulation in 3T3-L1 Adipocytes Without Cytotoxicity”
Manuscript ID: Biomedicines-3751311
Comments from Reviewer 1
Response to the comment 1:
We would like to sincerely thank the reviewer for the time and effort devoted to reviewing our manuscript. We are truly grateful for the constructive and insightful comments provided during the review process.
We are pleased to learn that the reviewer is now satisfied with our responses and has no further comments. We deeply appreciate the reviewer’s professional input and thoughtful evaluation, which have contributed greatly to improving the quality of our manuscript.
Thank you again for your kind support and consideration.
Once again, we deeply appreciate the Editor and Reviewers for your time, efforts, and constructive feedback. Your thoughtful comments have greatly improved the clarity, depth, and scientific rigor of our manuscript. We hope the revised version meets your expectations and look forward to your further guidance.
Sincerely,
Prof. Tai-Hsin Tsai
Department of Neurosurgery, Kaohsiung Medical University Hospital
No. 100, Tzyou 1st Road, Sham-min District, Kaohsiung City, Taiwan
E-mail: teishin8@hotmail.com
Reviewer 2 Report
Comments and Suggestions for Authors
The authors conducted all the required modifications, and I have no more comments
Author Response
Department of Neurosurgery
Kaohsiung Medical University Hospital
Kaohsiung, Taiwan
Jun 9th, 2025
To the Editor and Reviewers,
Manuscript Title: “Cucurbitacin E Inhibits Adipogenesis and Lipid Accumulation in 3T3-L1 Adipocytes Without Cytotoxicity”
Manuscript ID: Biomedicines-3751311
Comments from Reviewer 2:
Response to the comment :
We would like to sincerely thank the reviewer for the time and effort devoted to reviewing our manuscript. We are truly grateful for the constructive and insightful comments provided during the review process.
We are pleased to learn that the reviewer is now satisfied with our responses and has no further comments. We deeply appreciate the reviewer’s professional input and thoughtful evaluation, which have contributed greatly to improving the quality of our manuscript.
Once again, we deeply appreciate the Editor and Reviewers for your time, efforts, and constructive feedback. Your thoughtful comments have greatly improved the clarity, depth, and scientific rigor of our manuscript. We hope the revised version meets your expectations and look forward to your further guidance.
Sincerely,
Prof. Tai-Hsin Tsai
Department of Neurosurgery, Kaohsiung Medical University Hospital
No. 100, Tzyou 1st Road, Sham-min District, Kaohsiung City, Taiwan
E-mail: teishin8@hotmail.com
Reviewer 3 Report
Comments and Suggestions for Authors
The authors responded adequately and professionally to all the comments and made subsequent improvement.
I wish only to check if the sentence from the abstract « Additional molecular analyses, such as western blotting and RT-PCR, were used to examine the expression of key adipogenic markers», since they omitted those results in the final version.
Author Response
Department of Neurosurgery
Kaohsiung Medical University Hospital
Kaohsiung, Taiwan
Jun 9th, 2025
To the Editor and Reviewers,
Manuscript Title: “Cucurbitacin E Inhibits Adipogenesis and Lipid Accumulation in 3T3-L1 Adipocytes Without Cytotoxicity”
Manuscript ID: Biomedicines-3751311
Comments from Reviewer 3:
The authors responded adequately and professionally to all the comments and made subsequent improvement.
I wish only to check if the sentence from the abstract « Additional molecular analyses, such as western blotting and RT-PCR, were used to examine the expression of key adipogenic markers», since they omitted those results in the final version.
Response to the comment :
Thank you for your thoughtful comment and kind recognition of our revisions.
You are absolutely correct—since the Western blotting and RT-PCR results were removed from the final version of the manuscript to improve focus and clarity, we have also revised the abstract accordingly.
The sentence in question has been removed to maintain consistency and avoid referencing results that are no longer presented in the current version of the manuscript.
We are pleased to learn that the reviewer is now satisfied with our responses and has no further comments. We deeply appreciate the reviewer’s professional input and thoughtful evaluation, which have contributed greatly to improving the quality of our manuscript.
Once again, we deeply appreciate the Editor and Reviewers for your time, efforts, and constructive feedback. Your thoughtful comments have greatly improved the clarity, depth, and scientific rigor of our manuscript. We hope the revised version meets your expectations and look forward to your further guidance..
Sincerely,
Prof. Tai-Hsin Tsai
Department of Neurosurgery, Kaohsiung Medical University Hospital
No. 100, Tzyou 1st Road, Sham-min District, Kaohsiung City, Taiwan
E-mail: teishin8@hotmail.com